# STC-ViT: Spatio Temporal Continuous Vision Transformer for Weather Forecasting

## Abstract

Operational weather forecasting system relies on computationally expensive physics-based models. Recently, transformer based models have shown remarkable potential in weather forecasting achieving state-of-the-art results. However, transformers are discrete and physics-agnostic models which limit their ability to learn the continuous spatio-temporal features of the dynamical weather system. We address this issue with **STC-ViT**, a Spatio-Temporal Continuous Vision Transformer for weather forecasting. STC-ViT incorporates the continuous time Neural ODE layers with multi-head attention mechanism to learn the continuous weather evolution over time. The attention mechanism is encoded as a differentiable function in the transformer architecture to model the complex weather dynamics. Further, we define a customised physics informed loss for STC-ViT which penalize the model's predictions for deviating away from physical laws. We evaluate STC-ViT against operational Numerical Weather Prediction (NWP) model and several deep learning based weather forecasting models. STC-ViT, trained on 1.5-degree 6-hourly data, demonstrates competitive performance compared to state-of-the-art data-driven models trained on higher-resolution data for global forecasting. Our code and checkpoints are available at https://anonymous.4open.science/r/STC-ViT-92C7

## 1    Introduction

Operational weather forecasting systems rely on physics based Numerical Weather Prediction (NWP) models. While highly accurate, these system tend to be extremely slow with aggregated approximation errors and requiring high computational resources (Palmer et al., 2005; Andersson, 2022).

Weather forecasting can be formulated as a continuously evolving physical phenomenon which is typically expressed as time-dependent partial differential equations (PDEs). These PDEs capture the dynamic evolution of atmospheric variables over time, enabling the simulation of complex weather patterns and phenomena. Encoding these physical priors leads to more interpretable and physically consistent forecasting models (Couairon et al., 2024). Recent works have shown that using deep learning models as surrogates to model complex multi-scale spatio-temporal phenomena can lead to training-efficient models (Gupta & Brandstetter, 2022). Vision Transformer by Dosovitskiy et al. (2020) has emerged one of such models (Lessig et al., 2023; Nguyen et al., 2023b).

However, transformers only process discrete sequences of input data and are fundamentally discrete in nature Fonseca et al. (2023) which limit their ability in modelling the continuous evolving dynamics of weather system. Additionally, these models ignore the fundamental physical laws of our atmosphere which is essential for faithful weather modelling. In this work, we address the **non-continuity** problem in Vision Transformer (ViT) which arise in weather forecasting systems where data is present in discrete form. The distinctive and continuously evolving characteristics of weather data pose significant challenge in producing accurate forecasts. Additionally, we penalise the model for not following physical dynamics in the form a customized physics informed loss function.

We build upon the core ideas of Neural Ordinary Differential Equation (NODE) by Chen et al. (2018) and ViT (Dosovitskiy et al., 2020). *We propose STC-ViT which leverages the continuous learning paradigm to*

*effectively learn the complex spatio-temporal changes even from weather data recorded at coarser resolution.* The idea is to parameterize the attention mechanism by converting it into a differentiable function. Continuous temporal attention is calculated sample-wise and combined with the patch wise spatial attention to learn the spatio-temporal mapping of weather variables in the embedding space of the vision transformer. Furthermore, we add derivation as a pre-processing step to prepare the discrete data for continuous model and explore the role of normalization in continuous modelling.

In this paper we focus on the following research question: *How to design a computationally efficient data-driven weather forecasting system that learns the continuous latent representation while respecting the fundamental laws of atmospheric physics.* In summary, our contributions are as follows:

1. We propose spatio-temporal continuous attention to effectively learn and interpolate the spatio-temporal information for weather forecasting.

2. We introduce derivation as a pre-processing step to ensure better feature extraction for continuous spatio-temporal models.

3. We introduce physical constraints in our model via soft penalties in the form of a custom physics informed loss based on three fundamental physical laws of thermodynamics, kinetic enrgy and potential energy.

4. We perform extensive experiments on both WeatherBench and WeatherBench 2 to show the competitive performance of STC-ViT against state-of-the-art forecasting models.

## 2 Background

### 2.1 Neural ODE

Neural ODEs are the continuous time models which learn the evolution of a system over time using Ordinary Differential Equations (ODE) (Chen et al., 2018). The key idea behind Neural ODE is to model the derivative of the hidden state using a neural network. Consider a hidden state $h(t)$ at time $t$ In a traditional neural network, the transformation from one layer to the next could be considered as moving from time $t$ to $t+1$. In Neural ODEs, instead of discrete steps, the change in $h(t)$ over time is defined by an ordinary differential equation parameterized by a neural network:

$$\frac{dh(t)}{dt} = f(h(t), t, \theta) \tag{1}$$

where $\frac{dh(t)}{dt}$ is the derivative of the hidden state with respect to time, $f$ is a neural network with parameters $\theta$ and $t$ is the time variable, allowing the dynamics of $h(t)$ to change with time.

**ResNets vs Neural ODEs** To see the connection between ResNets and Neural ODEs, consider a ResNet with layers updating the hidden state as:

$$h_{t+1} = h_t + f(h_t, \theta_t) \tag{2}$$

In the limit, as the number of layers goes to infinity and the updates become infinitesimally small, this equation resembles the Euler method for numerical ODE solving, where:

$$h(t + \triangle t) = h(t) + \triangle t \cdot f(h(t), t, \theta) \tag{3}$$

Reducing $\triangle t$ to an infinitesimally small value transforms the discrete updates into a continuous model described by the ODE given earlier. To compute the output of a Neural ODE, integration is used as a backpropagation technique. This is done using numerical ODE solvers, such as Euler, Runge-Kutta, or more sophisticated adaptive methods, which can efficiently handle the potentially complex dynamics encoded by $f$.

## 2.2 Physics Constrained Models

In weather and climate modeling, incorporating physical constraints ensures that the model adheres to the governing physical laws. These constraints can be added in two ways: hard constraints and soft constraints. **Hard Constraints:** For a given physical constraint $f(x) = 0$ where $f(x)$ is the governing physical law, a hard constraint would mean that the machine learning model's prediction $\hat{y}$ must always satisfy equation 4.

$$f(\hat{y}) = 0 \tag{4}$$

This constraint can be embedded into model's architecture as a constrained layer or optimizer.

**Soft Constraints:** Soft constraints adds a penalty term to the loss function that minimizes the violation of a physical law $f(\hat{y})$ as shown in equation 5

$$min_\theta = L(y, \hat{y}) + \alpha ||f(\hat{y})||^2 \tag{5}$$

where $\alpha$ controls the weight of the penalty for violating the physical constraint $f(\hat{y})$ and $||f(\hat{y})||^2$ measures the degree of violation (e.g., deviation from mass conservation).

## 3 Related Work

Integrated Forecasting System (IFS) ECMWF (2023) is the operational NWP based weather forecasting system which generates forecasts at a high resolution of 1km. IFS combines data assimilation and earth system model to generate accurate forecasts using high computing super computers. In contrast, data-driven methodologies have now outperformed NWP models with much less computational complexities.

WeatherBench by Rasp et al. (2020) provides a benchmark platform to evaluate data-driven systems for effective development of weather forecasting models. Current state-of-the-art data-driven forecasting models are mostly based on Graph Neural Networks (GNNs) and Transformers. Keisler (2022) implemented a message passing GNN based forecasting model which was further extended by GraphCast (Lam et al., 2023) which used multi-mesh GNN to achieve state-of-the-art results. FourCastNet (FCN) Kurth et al. (2023) used a combination of Fourier Neural Operator (FNO) and ViT and was reported to be 80,000 times faster than NWP models. Several more transformer based models emerged including Pangu-Weather (Bi et al., 2023), ClimaX (Nguyen et al., 2023a), FengWu (Chen et al., 2023a), FuXi (Chen et al., 2023b) and Stormer (Nguyen et al., 2023b) all showcased remarkable capabilities for short to medium range forecasting.

While being highly accurate and showcasing remarkable scaling capabilities, these models are discrete space-time models and do not account for the continuous dynamics of weather system. Recently, Verma et al. (2023) proposed ClimODE which used Neural ODE to incorporate a continuous-time process that models weather evolution and advection, enabling it to capture the dynamics of weather transport across the globe effectively. However it currently yields less precise forecast results compared to state-of-the-art models, offering significant potential for further enhancements. Further, Kochkov et al. (2023) proposed Neural GCM, which integrates a differentiable solver with neural networks resulting in physically consistent models.

## 4 Methodology

**Problem Formulation.** Consider a model $f$ receives weather data as input of the form $X^{N \times W \times H}$ at time $t$ and predicts the weather information at time $t + \triangle t$ as shown in equation 6 where $N$ is the number of weather variables such as temperature, wind speed, pressure, etc. and $H \times W$ refers to the spatial resolution of the variable.

The objective of the model is to learn the continuous temporal dependencies while accounting for spatial correlations within the $H \times W$ grid. Since the weather changes continuously over time, it is essential to capture the continuous change within the provided fixed time step data. The main aim of STC-ViT is to to learn the continuous latent representation of the weather data using Spatio-Temporal Continuous Attention and Neural ODEs. The evolution of the weather system from $t$ to $t + \triangle t$ can be represented as:

$$X(t + \triangle t) = f(X, t) \tag{6}$$

$$\frac{dX(t)}{dt} = f(X, t, \theta)$$

$$\int_t^{t+\triangle t} \frac{dX(t)}{dt} = \int_t^{t+\triangle t} f(X, t, \theta)$$

$$X(t + \triangle t) = [f(X, t, \theta)]_t^{t+\triangle t} \; \theta = \text{learnable model parameters}$$

**Notation:** Throughout the paper $t$ and $t-1$ is used to denote the information at current and previous time step respectively. $V_i(x, y, t)$ is used to denote weather variable with x and y dimensions at current time step t.

**Derivation as a pre-processing step.** Weather information is highly variational and complex at both temporal and spatial levels. Temporal derivatives of each weather variable are calculated to preserve weather dynamics and ensure better feature extraction from discretized data. We perform sample wise derivation at pixel level to capture the continuous changes in weather events over time as shown in equation 7.

$$\frac{dV(x, y, t)}{dt} = \frac{V(x, y, t) - V(x, y, t-1)}{\Delta t} \tag{7}$$

where $V(x, y, t)$ is pixel value of weather variable at time t and $V(x, y, t-1)$ is the pixel value of same variable at t-1.

### 4.1 Spatio-Temporal Continuous Vision Transformer

In the STC-ViT architecture, we enhance vision transformer-based weather forecasting by introducing Temporal Continuous Attention (TCA) and Spatial Attention (SA) mechanisms to capture dynamically evolving weather patterns. We build upon the variable tokenization and aggregation scheme proposed by (Nguyen et al., 2023a) followed by Continuous attention mechanism to enhance interpolated feature learning. The detailed architecture of STC-ViT is shown in figure 2.

**Temporal Continuous Attention (TCA).** We model the temporal dynamics with attention by incorporating derivatives (temporal changes) over time directly into the query, key, and value representations.

We formulate **Query (Q)** as the "current" time step's information and is designed to seek relevant patterns or changes reflecting the model's focus on upcoming changes equation 8.

$$Q_t = \frac{dV_t}{dt} \tag{8}$$

**Key (K)** models the "context" of prior time step states, which provide historical dynamics incorporating past time step derivatives equation 9.

$$K_{t-1} = f(V_{t-1}, \frac{dV}{dt}|_{t-1}) \tag{9}$$

**Value (V)** represents the updated information allowing the model to interpret them in the context of previous and current changes. This modification allows the model to learn the transitional changes from one time step to another which is important for capturing unprecedented changes in weather.

To compute TCA and capture the temporal change for the same variable, we calculate the attention between $Q_{t,i}$ and $K_{t-1,i}$ focusing on the temporal evolution. The attention mechanism to capture the temporal continuity between time steps $t$ and $t-1$ is given by:

$$\frac{d}{dt}(Q_{t,i} \cdot K_{t-1,i}) = Q_{t,i}\frac{d(K_{t-1,i})}{dt} + K_{t-1,i}\frac{d(Q_{t,i})}{dt} \tag{10}$$

$$Y_i = Q_{t,i}\frac{d(K_{t-1,i})}{dt} + K_{t-1,i}\frac{d(Q_{t,i})}{dt} \tag{11}$$

$$Att(Q_{t,i}, K_{t-1,i}) = Softmax(\frac{Y_i}{\sqrt{d_k}}) \tag{12}$$

$$TCA_{t,t-1} = \sum_{k=1}^{N} Att(Q_{t,i}, K_{t-1,i}) \cdot \mathbf{V}_{t-1,i} \tag{13}$$

The resulting output represents the attention weighted sum of values for similar variables across input samples at time $t_0$ and $t_1$. This approach models the dynamics of each variable independently over time, allowing the model to capture temporal dependencies and changes effectively.

**Spatial Attention (SA).** For each variable i, we calculate the attention between spatial positions $(x, y)$ and $(x', y')$ given by equation 15:

$$Attention(Q_{x,y,i}, K_{x',y',i}) = softmax(\frac{Q_{x,y,i} \cdot K_{x',y',i}}{\sqrt{d_k}}) \tag{14}$$

$$SA_{t,i} = \sum_{x',y'} Attention(Q_{x,y,i}, K_{x',y',i}) \cdot \mathbf{V}_{x',y',i} \tag{15}$$

where Q, K are matrices formed from all queries, keys, and values, respectively, $d_k$ is the dimensionality of the keys and queries, and the division by $\sqrt{d_k}$ is a scaling factor to deal with gradient vanishing problems during training. Use the spatial attention weights to obtain the spatially-enhanced representation for each variable i by weighting the values:

**Concatenation and Fusion.** We concatenate the outputs of the TCA and SA along the feature dimension (as shown in equation 16) to provide a comprehensive representation, considering both temporal and spatial dependencies. This dual attention approach allows the Vision Transformer to effectively model complex spatio-temporal dependencies in the weather reanalysis data. It captures both the temporal evolution of each variable over time and the spatial interactions within each time step, enhancing the model's ability to understand the continuous, spatiotemporal dynamics of the data. The outputs of temporal and spatial attention are concatenated and projected through an output layer which forms the input for the Neural ODE component.

$$Attention(h) = concat(TCA_{t,i}, SA_{t,i}) \tag{16}$$

**Neural ODE Integration.** To enable continuous transformations, the output of the attention mechanism is treated as the initial state of a Neural ODE. The evolution of this state is governed by equation 17:

$$\frac{dh(t)}{dt} = f(h(t), t, \theta) \tag{17}$$

where $f$ is a learnable function parameterized as a multi-layer perceptron (MLP). The architecture of Neural ODE layer is shown in Figure 1

The Neural ODE evolves the input state $h(t)$ continuously between $t = 0$ and $t = 1$ as equation 18. The integration is performed numerically using the odeint function with Runge-Kutta (rk4)Runge (1895) numerical ODE solver, which evaluates $f(h(t), t, \theta)$ multiple times for precision.

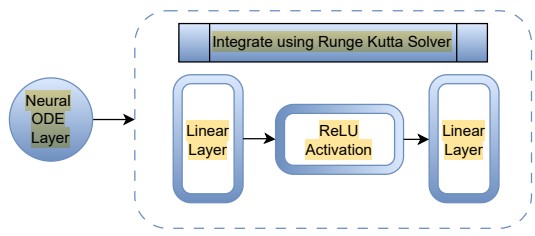

$$h(t_1) = h(t_0) + \int_{t_0}^{t_1} f(h(t), t, \theta)dt \tag{18}$$

The following equation 19 provides output for each lead time:

Figure 1: Neural ODE layer architecture

$$h_{n+1} = h_n + ODEBlock(LayerNorm(Attention(h_n))) + ODEBlock(LayerNorm(MLP(h_n))) \tag{19}$$

The model leverages Neural ODEs to capture smoother, continuous transformations. While this formulation aligns with continuous-depth modeling principles, the practical implementation focuses on fixed-interval evaluations rather than arbitrary time points.

### 4.2 Latitude Weighted Physics Informed Loss Function

We use latitude weighted mean squared error to compute loss for the predicted variables.

$$L_{lat-weight} = \frac{1}{N \times H \times W} \sum_{n=1}^{N} \sum_{i=1}^{H} \sum_{j=1}^{W} (L_i)(\hat{X}_{t+\Delta t}^{n,i,j} - X_{t+\Delta t}^{n,i,j}) \tag{20}$$

where $L_i$ accounts for Latitude weights:

$$L(i) = \frac{\cos(lat(i))}{\frac{1}{H} \sum_{i'=1}^{H} \cos(lat(i'))}$$

Additionally, we account for three fundamental physical laws i.e. Potential Energy, Kinetic Energy and Thermodynamic Balance in our loss function. Kinetic energy in atmospheric science refers to the energy associated with the motion of air masses. Geopotential is used in meteorology to express the potential energy of an air parcel in the Earth's gravitational field. Thermodynamic balance equation describes the evolution of temperature in an air parcel due to processes like heat addition and pressure changes. These fundamental principles, coupled with wind components and thermodynamic variables like temperature and geopotential, form the core of atmospheric dynamics used in climate and weather models.

$$L_{kinetic} = \left| \frac{1}{2}(u_{pred}^2 + v_{pred}^2) - \frac{1}{2}(u_{true}^2 + v_{true}^2) \right| \tag{21}$$

where u is the eastward wind component (m/s) and v is the northward wind component (m/s).

$$L_{potential} = |g \cdot z_{pred} - g \cdot z_{true}| \tag{22}$$

where g is the gravitational acceleration constant i.e. 9.81 and z geopotential $m^2/s^2$. Since z already reflects the potential energy, here it is used as a weighting mechanism to ensure that the loss calculation emphasizes discrepancies in regions with higher potential energy, thereby aligning the predictions more closely with the physical significance of geopotential.

$$L_{thermo} = \left| \frac{dT}{dt} + u\frac{dT}{dx} + v\frac{dT}{dy} \right| \tag{23}$$

where $\frac{dT}{dt}$ is the temporal change of temperature (model output), $u\frac{dT}{dx}, v\frac{dT}{dy}$ represent the advection of temperature by wind in the x and y directions, respectively.

$$Loss_{physics} = L_{lat-weight} + \alpha \times L_{kinetic} + \beta \times L_{potential} + \gamma \times L_{thermo} \tag{24}$$

where $\alpha$, $\beta$ and $\gamma$ are the weight factors for physics based loss. The resulting the loss function is inspired by physical principles, however it does not enforce strict physical laws but instead guides the model toward outputs that are consistent with physical expectations.

## 5 Experiments and Results

### 5.1 Dataset

We train STC-ViT on ERA5 dataset Hersbach et al. (2018; 2020) provided by the European Center for Medium-Range Weather Forecasting (ECMWF). We compare STC-ViT against several weather forecasting models by training it at two different resolutions of 5.625° (32 x 64 grid points) provided by WeatherBench (Rasp et al., 2020) and 1.5° (121 x 240 grid points) provided by WeatherBench2 (Rasp et al., 2024).

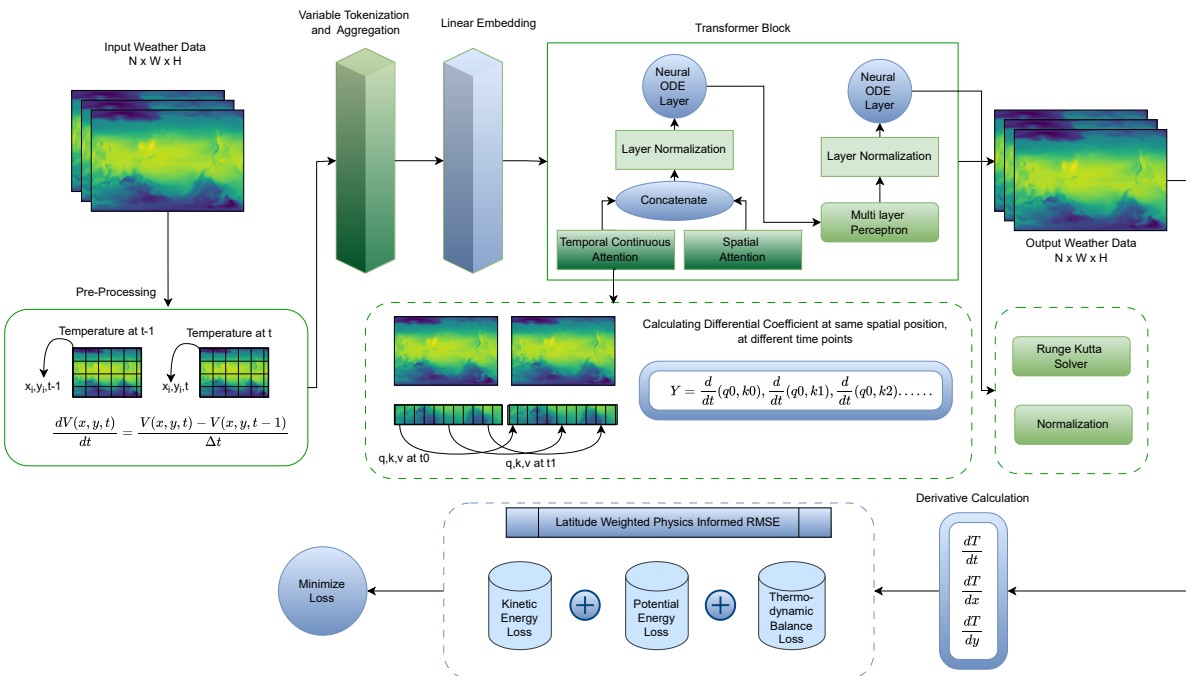

Figure 2: Overall prediction pipeline of **STC-ViT** as the model receives spatio-temporal weather information and passed through pre-derivation and normalization steps and input to transformer encoder where TCA and SA learns the continuous weather features and outputs the prediction

## 5.2 Training Details.

We consider weather forecasting as a continuous spatio temporal forecasting problem i.e a tensor of shape $N \times H \times W$ at time $t$ is fed to the pre-processing layer of the model where it passes through pre-derivation to STC-ViT and outputs a tensor of $N' \times H' \times W'$ at future time step $t + \triangle t$. Complete training details of the model are given in the in Appendix A.

## 5.3 WeatherBench

We train STC-ViT on hourly data with following set of variables: Land Sea Mask (LSM), Orography, 10-meter U and V wind components (U10 and V10) and 2-meter temperature (T2m) in addition to 6 atmospheric variables: geopotential (Z), temperature (T), U and V wind components, specific humidity (Q) and relative humidity (R) at 7 pressure levels: 50 250 500 600 700 850 925. We use data from 1979-2015 for training, 2016 for validation and 2017-2018 for testing phase. We compare STC-ViT with ClimaX, and ClimODE on ERA5 dataset at 5.625° resolution provided by WeatherBench (Rasp et al., 2020). To ensure fairness, we retrained ClimaX from scratch without any pre-training.

STC-Vit outperforms ClimaX and ClimODE at all lead times which shows that replacing regular attention with continuous attention in ViT architecture derives improved feature extraction by mapping the changes occurring between successive time steps. Additionally, enforcing physical constraints in the model lead to improved prediction scores. The RMSE and ACC results are shown in Table 1.

## 5.4 WeatherBench2

To keep training consistent with WeatherBench 2, we utilize the training data from 1979 to 2018, validation data from 2019, and test data from 2020 year. We train STC-ViT on 6 hourly data for following variables:

Table 1: RMSE and ACC results of $STC_{5.6}$ compared against ClimODE (ODE based) and ClimaX (non-pretrained) trained on ERA5 at 5.625° resolution

| Variable | Lead Time (hrs.) | RMSE (Lower is better) | | | ACC (Higher is better) | | |
|---|---|---|---|---|---|---|---|
| | | STC-ViT | ClimODE | ClimaX | STC-ViT | ClimODE | ClimaX |
| z500 ($m^2\backslash s^2$) | 6 | **78.36** | 102.2 | 249 | **0.99** | 0.99 | 0.97 |
| | 12 | **99.42** | 132.7 | 265.3 | **0.99** | 0.99 | 0.96 |
| | 18 | **117.16** | 163 | 319.8 | **0.99** | 0.98 | 0.95 |
| | 24 | **141.45** | 193.4 | 455 | **0.98** | 0.98 | 0.89 |
| | 36 | **206.59** | 259.6 | 455 | **0.97** | 0.96 | 0.89 |
| U10 ($m\backslash s$) | 6 | **0.92** | 1.44 | 1.58 | **0.97** | 0.91 | 0.92 |
| | 12 | **1.11** | 1.80 | 1.96 | **0.96** | 0.89 | 0.88 |
| | 18 | **1.28** | 1.97 | 2.24 | **0.95** | 0.88 | 0.84 |
| | 24 | **1.46** | 2.00 | 2.49 | **0.93** | 0.87 | 0.80 |
| | 36 | **1.89** | 2.25 | 2.95 | **0.89** | 0.83 | 0.70 |
| V10 ($m\backslash s$) | 6 | **0.95** | 1.53 | 1.60 | **0.97** | 0.92 | 0.92 |
| | 12 | **1.15** | 1.81 | 1.97 | **0.96** | 0.89 | 0.88 |
| | 18 | **1.32** | 1.95 | 2.26 | **0.94** | 0.88 | 0.83 |
| | 24 | **1.50** | 2.02 | 2.48 | **0.93** | 0.86 | 0.80 |
| | 36 | **1.92** | 2.29 | 2.94 | **0.88** | 0.83 | 0.70 |
| T2m (K) | 6 | **0.87** | 1.20 | 2.02 | **0.98** | 0.97 | 0.92 |
| | 12 | **1.04** | 1.44 | 2.26 | **0.97** | 0.96 | 0.90 |
| | 18 | **1.13** | 1.42 | 2.45 | **0.97** | 0.96 | 0.88 |
| | 24 | **1.18** | 1.40 | 2.37 | **0.97** | 0.96 | 0.89 |
| | 36 | **1.42** | 1.70 | 2.85 | **0.96** | 0.94 | 0.84 |
| T850 (K) | 6 | **0.83** | 1.16 | 1.64 | **0.98** | 0.97 | 0.94 |
| | 12 | **0.99** | 1.32 | 1.77 | **0.97** | 0.96 | 0.93 |
| | 18 | **1.09** | 1.47 | 1.93 | **0.97** | 0.96 | 0.92 |
| | 24 | **1.19** | 1.55 | 2.17 | **0.97** | 0.95 | 0.90 |
| | 36 | **1.44** | 1.75 | 2.49 | **0.95** | 0.94 | 0.86 |
| u50 ($m\backslash s$) | 6 | **1.42** | 2.01 | 2.05 | **0.98** | 0.92 | 0.92 |
| | 12 | **1.75** | 2.56 | 2.68 | **0.98** | 0.91 | 0.90 |
| | 18 | **1.90** | 2.89 | 2.92 | **0.96** | 0.90 | 0.89 |
| | 24 | **2.06** | 2.97 | 3.01 | **0.96** | 0.90 | 0.89 |
| | 36 | **2.37** | 3.03 | 3.12 | **0.94** | 0.88 | 0.87 |
| v50 ($m\backslash s$) | 6 | **1.45** | 2.01 | 2.12 | **0.95** | 0.90 | 0.90 |
| | 12 | **1.73** | 2.20 | 2.32 | **0.94** | 0.89 | 0.88 |
| | 18 | **1.89** | 2.27 | 2.31 | **0.92** | 0.87 | 0.85 |
| | 24 | **2.00** | 2.35 | 2.41 | **0.91** | 0.87 | 0.84 |
| | 36 | **2.19** | 2.51 | 2.59 | **0.90** | 0.85 | 0.83 |

T2m, u10 and v10 wind components and mean sea-level pressure (MSLP) along with five atmospheric variables: geopotential height (Z), temperature (T), U and V wind components, and specific humidity (Q). These atmospheric variables are considered at 13 pressure levels: 50, 100, 150, 200, 250, 300, 400, 500, 600, 700, 850, 925, 1000 hPa. We also compare our results with both versions of IFS Andersson (2022), IFS-HRES which is state-of-the-art forecasting model at high resolution of 0.1° and IFS-ENS, ensemble version trained at 0.2°. Additionally, we compare STC-ViT against GraphCast and PanguWeather which are trained at a higher resolution of 0.25° and finally with Neural GCM trained at 0.7°.

STC-ViT shows competitive performance, outperforming Pangu and GraphCast particularly for geopotential, temperature and wind variables. This is due to the physics loss penalizing the model for not obeying physics and encouraging outputs that balance temperature evolution and respect thermodynamic laws by penalizing deviations from these energy fluxes. We find out that Neural GCM is the best performing model on atmospheric variables which could be due to the physics embedded in the dynamical core. We believe that STC-ViT could also benefit from high resolution training and physics directly incorporated in the model architecture which we will explore in future. The RMSE and ACC results are shown in Figures 3 and 5 respectively.

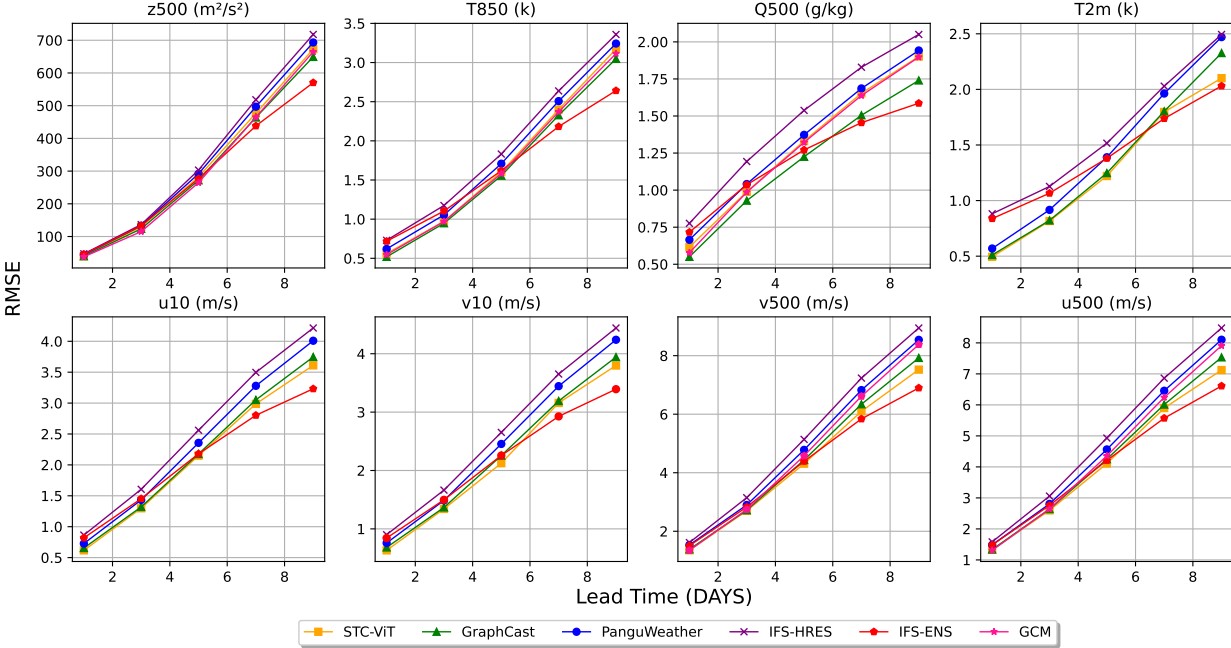

Figure 3: RMSE comparison of STC-ViT trained at 1.5° with GraphCast, PanguWeather, Neural GCM, IFS-ENS at 0.2° and IFS-HRES at 0.1° resolution data for lead times ranging from 1 to 10 days

## 5.5 Ablation Studies

### 5.5.1 STC-ViT Component Analysis

**Vanilla Vision Transformer** Vision Transformer has emerged as a powerful architecture which captures long-term dependencies better than any model. For this ablation study, we simply train a basic ViT architecture on ERA5 dataset. Compared with STC-ViT, ViT under performs for prediction accuracy showing the superiority of continuous models in weather forecasting systems.

**Vanilla Neural ODE Network.** Another ablation study is done using vanilla Neural ODE model. We replace the transformer with Neural ODE architecture as proposed in the original paper Chen et al. (2018). While Neural ODEs are computationally efficient, they only aid in interpolating the temporal irregularities and ignore the spatial continuity. This study proves that STC-ViT learns both spatio-temporal continuous features from the discrete data and is better at representing dynamical forecast systems.

**Continuous Attention.** To understand the importance of continuous attention, we remove the neural ODE layer and provide the output of attention mechanism to the basic feed forward network. While continuous attention only model does not outperform the STC-ViT, it provides better prediction accuracy than the traditional attention model.

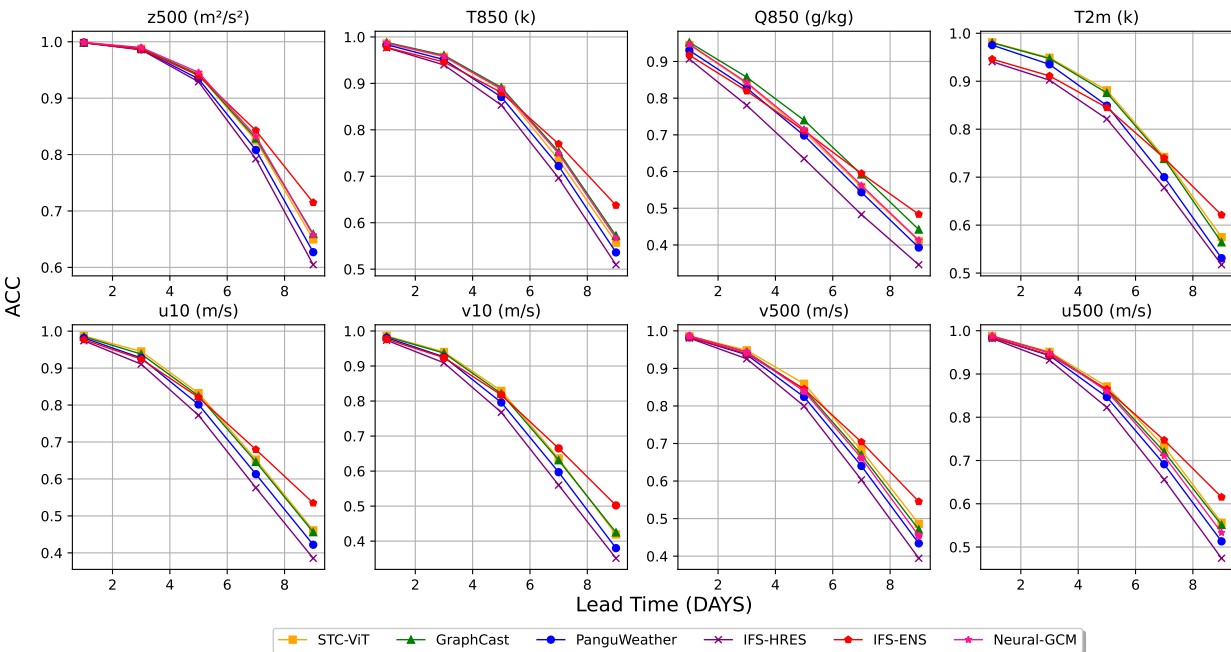

Figure 4: ACC comparison of STC-ViT trained at 1.5° with GraphCast, PanguWeather, Neural GCM, IFS-ENS at 0.2° and IFS-HRES at 0.1° resolution data for lead times ranging from 1 to 10 days

**Continuous Attention + Neural ODE layer.** We perform another ablation study to compare how well Neural ODE capture continuity in weather information when simply added as a layer in the transformer architecture. Neural ODE Chen et al. (2018) is a continuous depth neural network designed to learn continuous information in the process as well. For this study, we replace the continuous attention with vanilla attention and add a Neural ODE layer in the feed forward block. The feed forward solution is approximated using Runge-Kutta (RK) numerical solver.

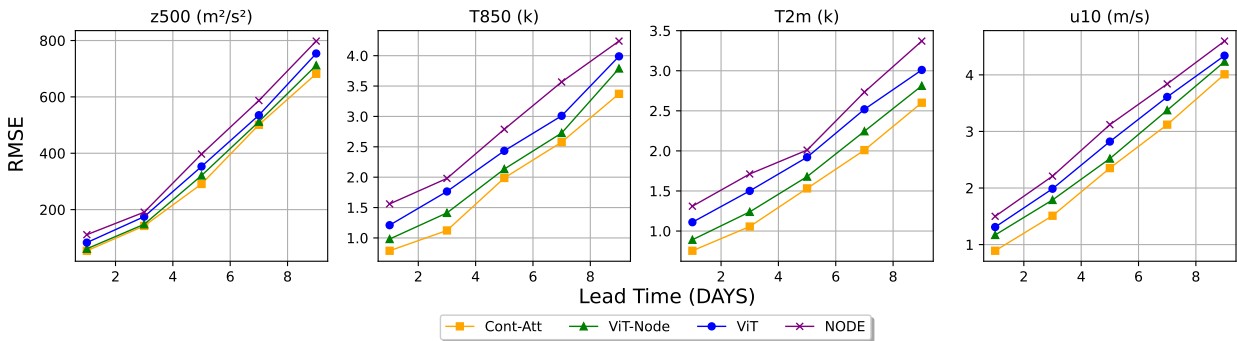

Figure 5: Ablation studies highlight the results for different components of STC-ViT individually and show that the continuous attention component alone performs better than all other even NODE. Thus combined with NODE outperforms state-of-the-art results

### 5.5.2 STC-ViT Loss Analysis

We further performed ablation studies to understand the effect of each loss function on the predictions. The results are shown in figure 6.

**Physics Uninformed - Latitude Weighted RMSE:** We trained STC-ViT solely using Latitude Weighted RMSE without including any physical loss terms. We observe that model's performance got worse without considering the physical laws in the loss function.

**Potential Energy Loss (PE):** To understand the role of PE loss, we assign the highest weight (0.8) to the potential energy loss function and keep the weights of Kinetic Energy and Thermodynamic loss to 0.1. We observe that it has the lowest influence on the predictions.

**Kinetic Energy Loss (KE):** We repeated the above study by assigning the highest weight (0.8) to KE loss and set the weights of PE and Thermodynamic loss equal to 0.1. The results suggest that its influence on the overall prediction accuracy is more as compared to Potential Energy Loss.

**Thermodynamic Loss:** Finally, we assign the highest weight to the Thermo Loss (0.8) while keeping the weights of PE and KE loss to negligible 0.1. We observe that Thermo loss has the highest influence on overall prediction accuracy resulting in the lowest RMSE as compared to all other loss terms.

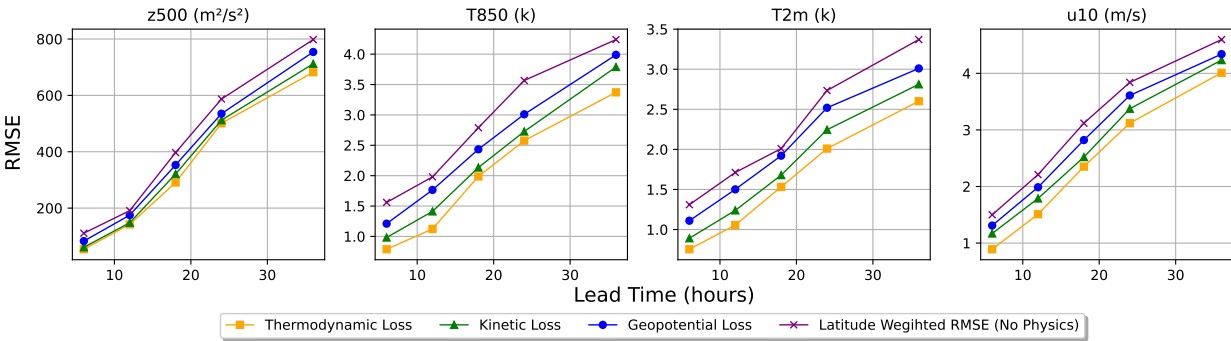

Figure 6: Ablation studies highlight the results for different loss terms of STC-ViT individually and show that the Thermodynamic Loss has the largest effect on the resulting RMSE.

## 6   Conclusion and Future Work

In this paper, we present STC-ViT, a novel technique designed to capture continuous dynamics of weather data while obeying the fundamental physical laws of Earth's atmosphere. STC-ViT achieves competitive results in weather forecasting which shows that vision transformers can model continuous nature of spatio-temporal dynamic systems with carefully designed attention mechanism.

While STC-ViT performs competitively with state-of-the-art data-driven weather forecasting models, it is important to address its limitations in weather forecasting systems. STC-ViT is inherently based on transformer architecture which has a limitation of having higher training times when scaled to higher resolutions as shown in Table 2. Further the deterministic nature of our approach does not account for uncertainties which can produce unrealistic results. Additionally, predictions on longer lead times result in blurry forecasts. Extending STC-ViT to a probabilistic model can be addressed in future works.

It is also important here to address the biases resulting from training on one dataset. The ERA5 dataset, while a robust and widely used weather dataset also has inherent limitations. Potential biases in data-sparse regions, challenges in representing local phenomena, and inconsistencies in observational continuity may impact model generalizability. To address this, future work will explore: data augmentation to synthetically increase dataset diversity. Integration with Diverse Datasets, such as regional or event-specific datasets, to mitigate geographical and climatic biases. Also, scaling STC-ViT to better accommodate the multi-modal high resolution training and evaluation presents an opportunity as future research. Finally, addressing the black-box problem of STC-ViT can shed light on model learning insights which is equally important for climate science community.

Table 2: Run-Time comparison against several data-driven weather forecasting models

| Model | Parameters | Training Time | Train Device |
|-------|-----------|---------------|--------------|
| PanguWeather | 256M | 64 days | 192 NVIDIA Tesla-V100 GPUs |
| GraphCast | 37M | 4 weeks | 32 TPUs |
| ViT (ClimaX-non pretrained) | 107M | 2 days | 8 V100 |
| STC-ViT (5.625/1.5 degree) | 98M | 2days/20days | 4 V100/2 A100 |

## Societal Impact

Our research focuses on modelling the continuous dynamics of weather forecasting system through the integration of deep learning (DL) techniques. The study shows that leveraging data-driven approaches can achieve significantly improved forecast results with compute efficient resources. The environmental benefits of compute-efficient forecasting systems are significant. Lowering the carbon footprint of computational processes contributes to global efforts in combating climate change. By integrating ML to improve accuracy while optimizing computational efficiency, we can create a sustainable and inclusive approach to weather forecasting that serves the global community more effectively. Additionally, training STC-ViT at higher resolution on more diverse datasets and focusing on high resolution regional forecasts as a future work can directly contribute to enhanced climate resilience, enabling societies to better anticipate and adapt to extreme weather events such as hurricanes, droughts, and floods. Further, extending the study to cater for longer lead times of few months and seasonal can help in disaster preparedness, as timely and precise forecasts allow governments and organizations to implement early warning systems, plan evacuations, and allocate resources effectively, thereby mitigating loss of life and property.

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

## A   Experiment Details

### A.1   Evaluation Metrics.

We used Root Mean Square Error (RMSE), Anomaly Correlation Coefficient (ACC) and Mean Absolute Error (MAE) to evaluate our model predictions. The formula used for RMSE is:

$$RMSE = \frac{1}{N} \sum_{k=1}^{N} \sqrt{\frac{1}{H \times W} \sum_{i=1}^{H} \sum_{j=1}^{W} L(i)(\hat{X}_{k,i,j} - X_{k,i,j})^2} \tag{25}$$

where $H \times W$ is the spatial resolution of the weather input and N is the number of total samples used for training or testing. L(i) is used to account for non-uniformity in grid cells.

$$ACC = \frac{\sum_{k,i,j} \hat{X}'_{k,i,j} X'_{k,i,j}}{\sqrt{\sum_{k,i,j} L(i)\hat{X}'^2_{k,i,j} \sum_{k,i,j} L(i)X'^2_{k,i,j}}} \tag{26}$$

Where $\hat{X}' = \hat{X}' - C$ and $X' = X' - C$ and C is the temporal mean of the entire test set $C = \frac{1}{N} \sum_k X$

$$MAE = \frac{1}{N} \sum_{k=1}^{N} \left| \frac{1}{H \times W} \sum_{i=1}^{H} \sum_{j=1}^{W} L(i)(\hat{X}_{k,i,j} - X_{k,i,j}) \right| \tag{27}$$

### A.2   Optimization

We use the AdamW optimizer Loshchilov & Hutter (2017) with a learning rate of 5e-5. We utilize a CosineAnnealing learning rate scheduler is adopted which progressively lowers the learning rate to zero following the warm-up period which is 10% of the total epochs. We train STC-ViT for 50 epochs for 5.625° and 100 epochs for 1.5° resolution data .

### A.3   Hyperparameters

### A.4   Normalization

In our experiments, we normalize all inputs during training and re-scale them to their original range before computing final predictions. We perform z-score normalization for every variable, at each atmospheric pressure level, we calculate the mean and standard deviation to standardize them to zero mean and unit variance.

### A.5   Software and Hardware Requirements

We use PyTorch Paszke et al. (2019), Pytorch Lightning Falcon (2019), torchdiffeq Chen et al. (2018), and xarray Hoyer & Hamman (2017) to implement our model. We use 2 NVIDIA DGX A100 devices with 80GB RAM for training STC-ViT at 1.5° and 4 NVIDIA Tesla Volta V100-SXM2-32GB for the training at resolution of 5.625°.

Table 3: Hyperparameters of STC-ViT

| Hyperparameters | Meaning | Value |
|---|---|---|
| p | Patch size | 2 |
| Heads | Number of continuous attention heads | 16 |
| Depth | Number of Transformer layers | 4 |
| Dimension | Hidden dimensions | 1024 |
| Dropout | Dropout rate | 0.1 |
| batch_size | Batch Size | 12 for 5.625°, 2 for 1.5° |
| $\beta 1$ | First Moment Decay Rate of AdamW optimizer | 0.9 |
| $\beta 2$ | Second Moment Decay Rate of AdamW optimizer | 0.999 |
| ES | Early stopping | True |
| ES rate | Early stopping tolerance | 10 |
| $\alpha$ | Kinetic Loss weight factor | 0.1-0.5 |
| $\beta$ | Potential Loss weight factor | 0.1-0.5 |
| $\gamma$ | Thermodynamic Loss weight factor | 0.8 |

# B  Results

## B.1  Quantitative Results

Table 4: RMSE for sub seasonal forecasts ranging from 2 weeks to 8 weeks

| Variable | 2 weeks | 4 weeks | 6 weeks | 8 weeks |
|---|---|---|---|---|
| z500 $(m^2/s^2)$ | 825.68 | 857.09 | 968.5 | 1068.5 |
| T2m (k) | 3.68 | 4.31 | 5.24 | 6.35 |
| T850 (k) | 3.67 | 4.05 | 4.82 | 5.54 |
| u10 (m/s) | 4.00 | 4.05 | 4.23 | 4.45 |
| v10 (m/s) | 4.07 | 4.13 | 4.28 | 4.42 |

Table 5: Mean Absolute Error (lower is better) of $STC_{5.6}$

| Variable | 6 hr | 12 hr | 18 hr | 1 day | 3 day | 6 day | 2 week | 4 week | 6 week |
|---|---|---|---|---|---|---|---|---|---|
| z500 $(m^2 \backslash s^2)$ | 60.33 | 72.33 | 84.79 | 98.67 | 351.34 | 519.87 | 529.97 | 557.38 | 722.47 |
| u10 $(m \backslash s)$ | 0.65 | 0.79 | 0.88 | 1.00 | 2.30 | 2.83 | 2.85 | 2.92 | 3.11 |
| v10 $(m \backslash s)$ | 0.67 | 0.81 | 0.91 | 1.03 | 2.36 | 2.89 | 2.92 | 2.99 | 3.16 |
| T2m (K) | 0.61 | 0.72 | 0.78 | 0.81 | 1.44 | 2.26 | 2.39 | 2.77 | 4.15 |
| T850 (K) | 0.60 | 0.726 | 0.79 | 0.86 | 1.81 | 2.53 | 2.63 | 2.90 | 3.96 |

**Spatial Resolution Analysis**

When analyzing inference RMSE across different spatial resolutions as shown in figure 7, we observe that STC-ViT trained at 1.5-degree spatial resolution offers significant improved prediction accuracy, however, as the lead time increases the gap between the RMSE values decreases which shows that models trained on 5.625 degree can offer better insights into longer sub seasonal forecasts and STC-ViT trained on 1.5 degree can be used to generate localized high resolution forecast for nowcasting and extreme event forecasting.

In conclusion, there is definitely a trade-off between resolution and accuracy across different forecasting scenarios and computational constraints.

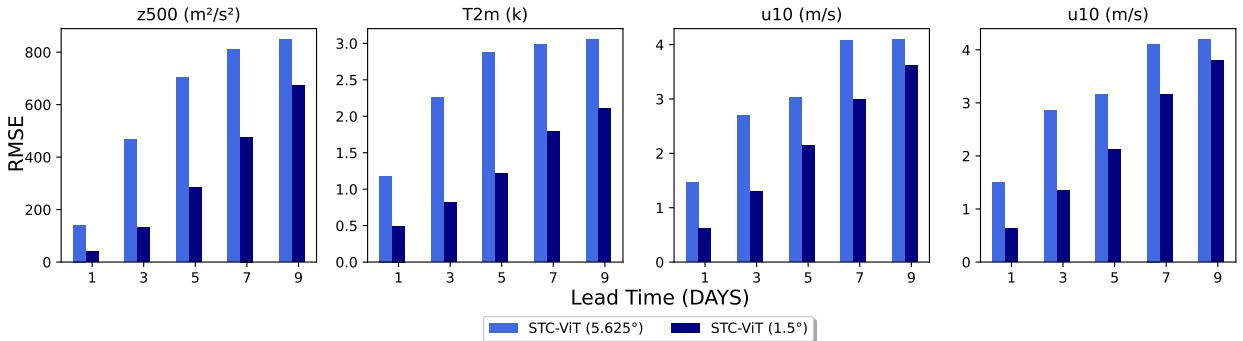

Figure 7: RMSE comparison between STC-ViT trained on 5.625 and 1.5 degree resolution

## B.2 Qualitative Results

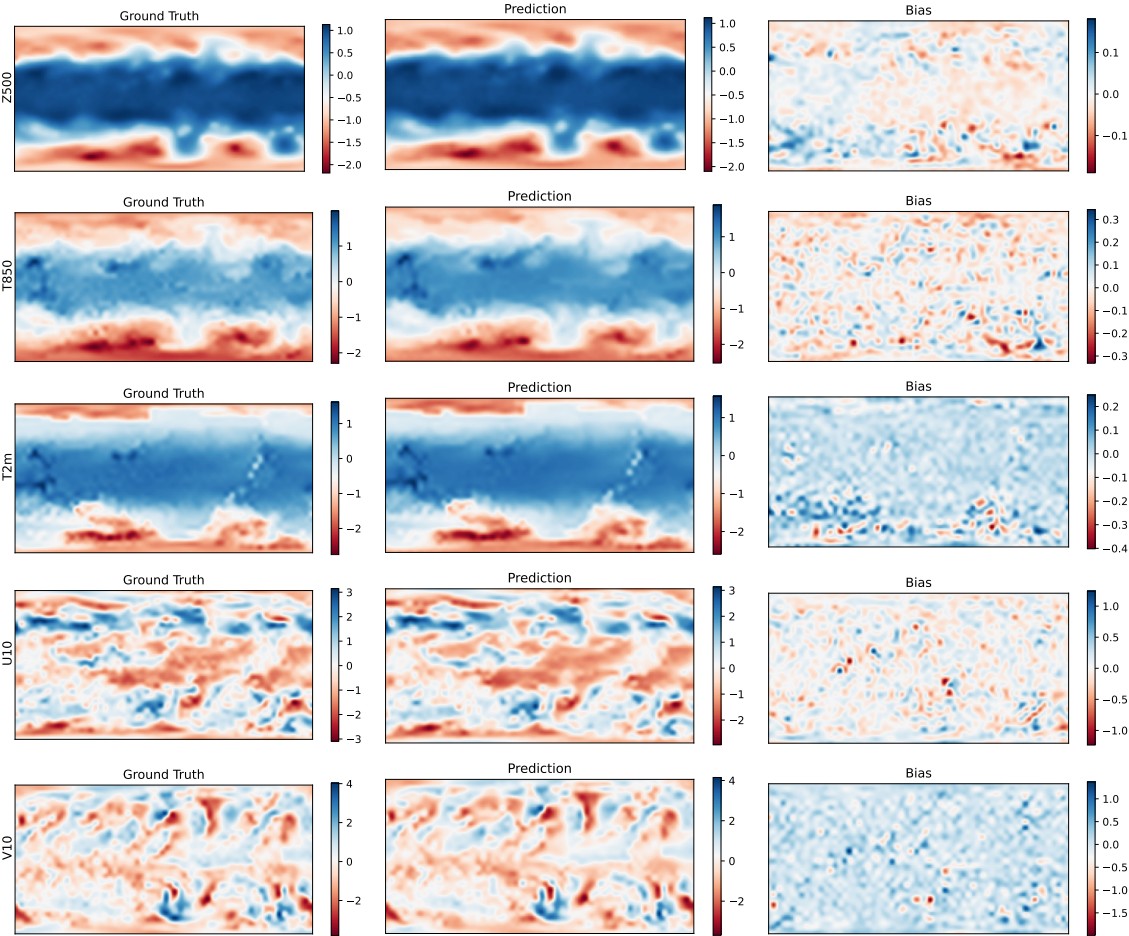

Figure 8: 12hr forecast results of STC-ViT

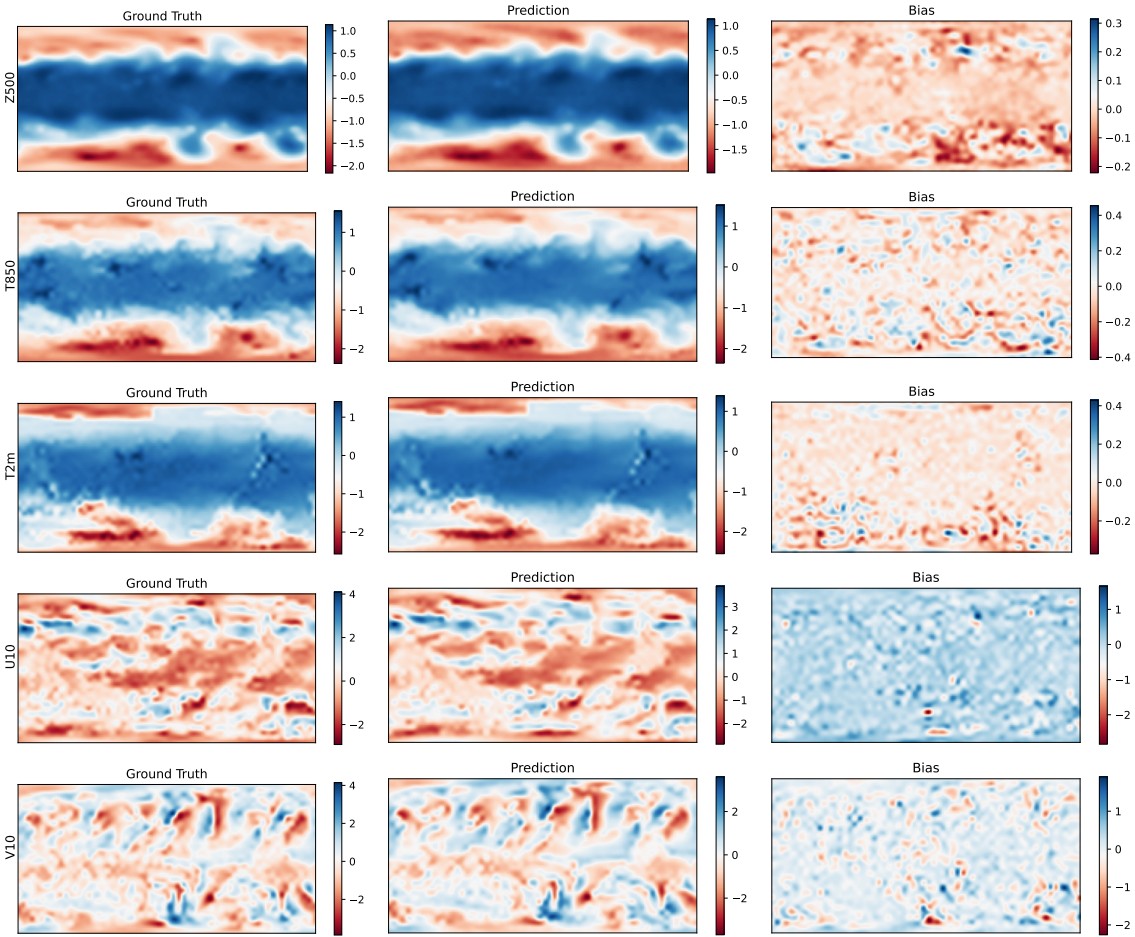

Figure 9: 1 day forecast results of STC-ViT

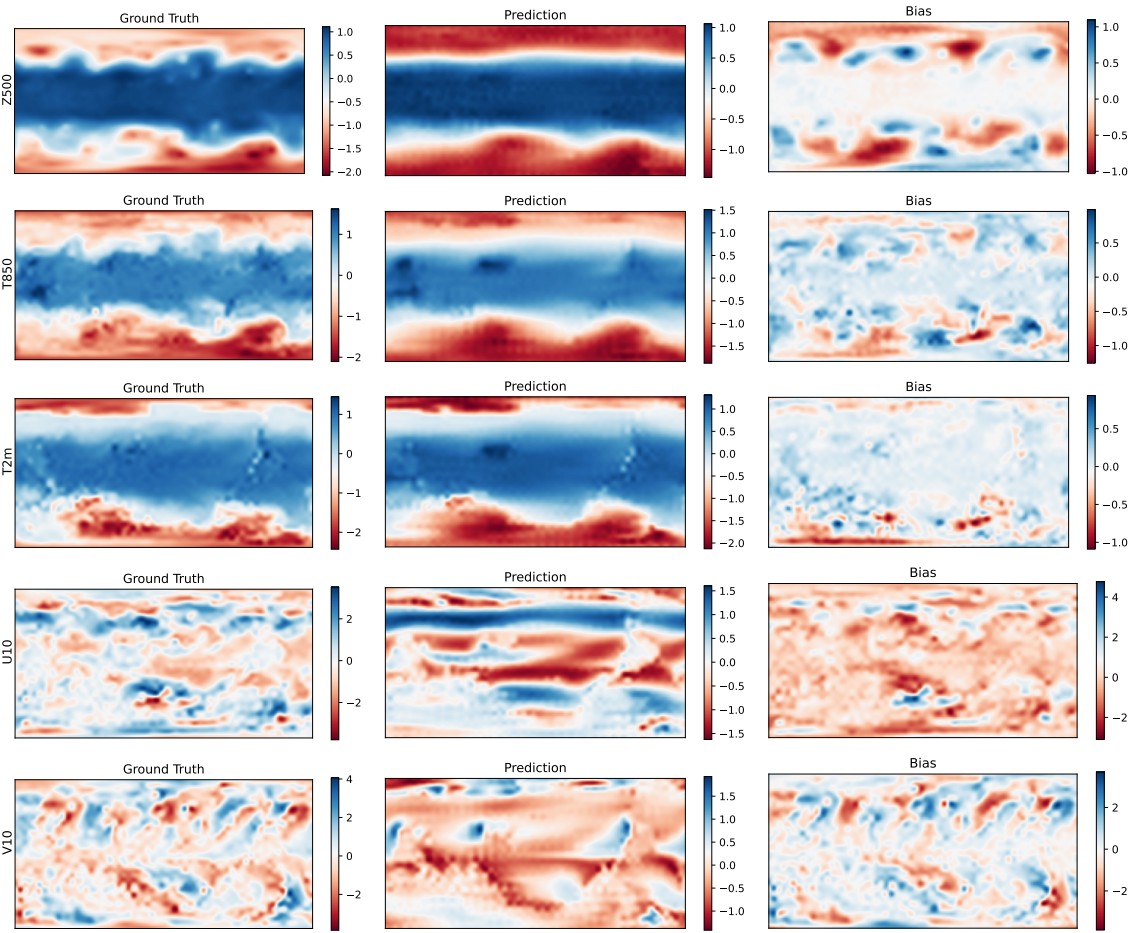

Figure 10: 2 weeks forecast results of STC-ViT

