# OpenReview forum: "STC-ViT: Spatio Temporal Continuous Vision Transformer for Weather Forecasting"
_TMLR — Rejected by TMLR_

### Review · Reviewer_1yfU · 2024-11-21

**Summary Of Contributions:**

STC-ViT integrates Neural ODE layers with a multi-headed attention mechanism to model the continuous evolution of weather conditions. The model incorporates a physics-informed loss function to penalize deviations from fundamental physical laws. Experimental results demonstrate that STC-ViT, trained on low-resolution (1.5°) data, achieves performance comparable to state-of-the-art models trained on higher-resolution data.

**Audience:**

Yes

**Claims And Evidence:**

Yes

**Requested Changes:**

**Questions**

- Could the authors provide an ablation study on the impact of the Physics-Informed RMSE loss?
- Do the authors have any insights into how inference RMSE varies when STC-ViT is trained at different spatial resolutions?

**Strengths And Weaknesses:**

**Strengths**

- The paper is clearly written and easy to follow.
- To the best of my knowledge, the integration of Neural ODEs with vision transformers to capture continuous spatiotemporal dynamics is novel and well-motivated.
- The ablation studies are thorough and provide valuable insights into the contributions of each key component in the proposed architecture.
- STC-ViT achieves competitive performance compared to both state-of-the-art deep learning models and NWP methods.

**Weaknesses**

- The model is trained on low-resolution data, which may limit its applicability for high-resolution forecasts requiring finer spatial granularity.
- As noted in the limitations section, the model does not account for uncertainty which is a critical aspect of weather forecasting.

---

> ### Author Response · Authors · 2024-12-05
> **Response to Reviewer 1yfU**
>
> We greatly appreciate your feedback on our work. Below, we address the weakness followed by the requested changes in detail.
> ### Weaknesses:
>
> **Weakness 1:** The model is trained on low-resolution data, which may limit its applicability for high-resolution forecasts requiring finer spatial granularity.
> - The choice of 5-degree and 1.5-degree resolution data was motivated by the computational constraints of model training on a higher resolution spatial scale. However, proposed attention mechanism facilitates learning temporal dynamics, which remain consistent across resolutions. Further, we acknowledge this weakness in limitations section as further experiments are required to quantify the model's performance on higher-resolution data and compare it to traditional approaches for high-resolution forecasting.
>
> **Weakness 2:** As noted in the limitations section, the model does not account for uncertainty which is a critical aspect of weather forecasting.
> - The primary objective of the current work was to establish the foundational architecture, focusing on capturing spatiotemporal dynamics efficiently. As a future work, we aim to extend the study to integrate uncertainty estimations to make the model’s predictions more trustworthy
>
> ### Requested Changes:
> - Ablation Studies on Physics Informed RMSE have now been included in the revised version (Refer to the section 5.5.2 and figure 6). We observe in our study that Thermodynamic law has the highest impact out of 3 physical laws and considering all 3 of them improves the results significantly.
> - We have added a discussion on resolution scaling and how it affects the RMSE in appendix (Refer to the section B in appendix) . We observe that increasing the spatial resolution significantly impacts the resulting RMSE. Further insights can be provided into this by training STC-ViT on a higher resolution of 0.25 degree which could be explored as a future work.

---

> > ### Comment · Reviewer_1yfU · 2024-12-14
> > **Response**
> >
> > Thank you for your responses, my concerns have been fully addressed.

---

### Review · Reviewer_C1mc · 2024-11-23

**Summary Of Contributions:**

The submission presents a significant advancement in the field of weather forecasting through the introduction of the Spatio-Temporal Continuous Vision Transformer! This innovative model addresses the limitations of traditional transformer architectures, which are often discrete and physics-agnostic, by incorporating continuous-time Neural Ordinary Differential Equations (ODE) layers alongside a multi-head attention mechanism! This integration allows STC-ViT to effectively learn and interpolate the complex spatio-temporal dynamic inherent in weather systems, thereby enhancing the model's ability to capture the continuous evolution of weather phenomena over time!

One key contribution of this work is the development of a continuous spatio-temporal attention mechanism, which is parameterized as a differentiable function! This novel approach facilitates the extraction of meaningful features from weather data, even when recorded at coarser resolutions! Additionally, the authors introduce derivation as a pre-processing step, which further improves feature extraction for continuous spatio-temporal models! this methodological innovation is particularly noteworthy, as it demonstrates a clear pathway for enhancing the performance of data-driven weather forecasting systems!

Moreover, the submission emphasizes the importance of incorporating physical constraints into the modeling process. By implementing a customized physics-informed loss function that penalizes deviations from fundamental physical laws—specifically those related to thermodynamics, kinetic energy, and potential energy—the authors ensure that the STC-ViT model adheres to the governing principles of atmospheric physics. This aspect not only enhances the model's reliability but also bridges the gap between data-driven approaches and established physical theories, thereby contributing to a more robust understanding of weather dynamics!

The empirical evaluation of STC-ViT against operational Numerical Weather Prediction (NWP) models and other deep learning-based forecasting systems demonstrates its competitive performance and computational efficiency. Trained on 1.5-degree, 6-hourly data, STC-ViT shows promise in achieving state-of-the-art results while utilizing fewer computational resources compared to higher-resolution models. This aspect is particularly relevant in the context of growing concerns about the environmental impact of computational processes, as the authors highlight the potential for reduced carbon footprints in weather forecasting!

**Audience:**

Yes

**Broader Impact Concerns:**

From my understanding, It is crucial to consider the ethical implications associated with the deployment and application of such models, especially in the context of societal impact and environmental sustainability!

One primary concern is the potential for biased predictions arising from the training data used to develop STC-ViT. If the model is trained on historical weather data that contains systemic biases - such as the underrepresentation of certain geographical areas or climatic conditions - there is a risk that the generated forecasts may not be equitable or accurate across diverse regions. This could disproportionately affect vulnerable communities that rely heavily on accurate weather predictions for agriculture, disaster preparedness, and public safety. Therefore, the authors need to clarify how they plan to ensure that the training data is representative and inclusive, thereby mitigating the risk of biased outcomes!

Additionally, while the computational efficiency of STC-ViT may reduce resource consumption, it raises concerns about the environmental impact of large-scale model training and deployment. The energy consumption associated with training deep learning models can contribute to carbon emissions, which could undermine the overarching goal of addressing climate change. The authors should provide a clear analysis of the model's carbon footprint compared to traditional forecasting methods and discuss strategies for minimizing environmental impact, such as optimizing algorithms for energy efficiency or using renewable energy sources in the computational process!

Furthermore, the potential misuse of advanced weather forecasting models should be considered. As prediction accuracy improves, there is a risk that such models could be exploited for malicious purposes, such as manipulating market behavior in agriculture or insurance sectors based on forecasted weather events. The authors should reflect on these ethical considerations and propose guidelines or frameworks for the responsible use of their technology to prevent potential exploitation!

Finally, the broader implications of the research on public policy and community resilience should be addressed. As STC-ViT demonstrates improved forecasting capabilities, it may influence decision-making processes at various levels, from local governments to international organizations. The authors should explore how their work can contribute to equitable policy development that prioritizes the needs of marginalized communities and enhances overall societal resilience to climate-related challenges!

**Claims And Evidence:**

Yes

**Requested Changes:**

To enhance the work, a detailed analysis of the methods and results highlights several proposed adjustments. These are categorized based on their critical importance for securing acceptance and their potential to strengthen the overall work!

Firstly, it's crucial to refine the methodology section by offering a more comprehensive explanation of the Neural ODE layers integrated within STC-ViT. Although the current submission mentions the use of Neural ODE, a deeper exploration of how these layers interact with the multi-head attention mechanism is needed. This should include a precise mathematical formulation, a step-by-step description of the training process, and an explanation of how the Neural ODE layers contribute to capturing the continuous dynamics of weather data. Clarifying these aspects is essential for ensuring that reviewers fully appreciate the innovative elements of the methodology, which is critical for acceptance! Secondly, the results section should provide a broader evaluation of the model's performance across multiple metrics. Currently, the submission focuses primarily on RMSE as a performance indicator. To give a more comprehensive view of the model's capabilities, it is vital to include additional metrics such as Mean Absolute Error (MAE), correlation coefficients, and skill scores. Additionally, contextualizing results across different weather phenomena (e.g., extreme weather events, seasonal variations) would demonstrate the model's robustness and adaptability across diverse conditions, thereby strengthening the submission!

In addition, conducting a thorough ablation study to systematically assess the contributions of each STC-ViT component is important. While some preliminary results are included, a more structured approach that isolates the effects of the continuous attention mechanism, Neural ODE layers, and the physics-informed loss function would offer clearer insights into the model's performance. This step is crucial for validating the design choices and for establishing the significance of each component in achieving the reported results! Furthermore, the authors should discuss the computational efficiency of STC-ViT compared to traditional NWP models. While the submission claims improved computational efficiency, providing quantitative comparisons in terms of training time, inference speed, and resource usage would substantiate these claims. This adjustment is essential for demonstrating the practical applicability of STC-ViT in operational settings, a key factor for acceptance!

Including a discussion of the limitations of the current study would also be beneficial. Acknowledging potential drawbacks, such as the model's performance in specific geographical regions or under particular atmospheric conditions, would enhance the study's credibility. This section should also explore how these limitations may inform future research and model development. Although not critical for acceptance, this adjustment would significantly strengthen the submission's integrity! Lastly, the discussion on the societal impact of the work should be expanded. While the submission briefly mentions environmental benefits, a more detailed analysis of how STC-ViT can enhance climate resilience, disaster preparedness, and public safety would create a compelling narrative about the broader implications of the research. This addition would not only enhance the work but also align it with current global priorities in climate science and policy.

**Strengths And Weaknesses:**

STC-ViT for weather forecasting showcases several exceptionally strong aspects that significantly enhance its contribution to the field! One of the most compelling strengths is the innovative integration of continuous-time Neural ODE with a multi-head attention mechanism! This methodological advancement allows the model to effectively capture the continuous dynamics of weather systems, addressing a critical limitation of traditional transformer architectures, which are inherently discrete! By building on the foundational work of the related works, the authors have successfully created a framework that not only leverages the strengths of these prior models but also extends their applicability to the complex and evolving nature of atmospheric phenomena!

Another notable strength of this work is the incorporation of a physics-informed loss function! This aspect is particularly significant, as it aligns the model's predictions with fundamental physical laws governing atmospheric dynamics, thereby enhancing the interpretability and reliability of the forecasts! The authors' approach resonates with the findings of Couairon et al. (2024), which emphasize the importance of integrating physical priors into machine learning models for improved forecasting accuracy. By penalizing deviations from these physical laws, the authors ensure that STC-ViT remains grounded in the principles of meteorology, a commendable and necessary step in the development of robust forecasting systems!

The empirical evaluation of STC-ViT against established operational Numerical Weather Prediction (NWP) models and other deep learning approaches further underscores the model's competitive performance. The authors provide a thorough analysis of the model's capabilities, demonstrating that STC-ViT achieves SOTA results while maintaining computational efficiency. This aspect is particularly relevant in the context of increasing concerns about the environmental impact of computational resources in weather forecasting, as highlighted by recent studies such as those by Gupta & Brandstetter (2022). The ability to deliver high-quality forecasts with reduced computational demands positions STC-ViT as a promising tool for operational applications.

However, despite these strengths, there are several areas that require attention to enhance the overall robustness and applicability of the work. One significant concern is the deterministic nature of the STC-ViT model, which does not account for model uncertainty. This limitation could lead to unrealistic forecasts, particularly in scenarios characterized by high variability or extreme weather events. To address this issue, the authors may consider exploring probabilistic modelling techniques, such as Bayesian neural networks or ensemble methods, which have been shown to effectively capture uncertainty in predictions. Incorporating such approaches could significantly improve the model's reliability and applicability in real-world forecasting scenarios!

Additionally, while the submission provides a solid foundation for the model's architecture and performance, there is a need for a more comprehensive discussion of the limitations and potential biases in the training data. The authors utilize the ERA5 dataset, which, while extensive, may still contain biases that could affect the model's generalizability. Acknowledging these limitations and discussing strategies for mitigating them, such as data augmentation or incorporating diverse datasets, would strengthen the submission and provide a more balanced perspective on the model's applicability across different geographical and climatic contexts!

Furthermore, the authors could enhance the interpretability of the model by providing deeper insights into the attention mechanisms employed within STC-ViT. While the continuous attention component is highlighted as a key feature, a more thorough exploration of how this attention mechanism operates in practice and its impact on the model's predictions would be beneficial. This could involve visualizations or case studies illustrating the model's decision-making process in specific forecasting scenarios, thereby providing a clearer understanding of how STC-ViT derives its predictions from the input data!

---

> ### Author Response · Authors · 2024-12-05
> **Response to Reviewer C1mc**
>
> We greatly appreciate your thorough feedback on our work. Below, we address the weakness followed by the requested changes in detail.
> ### Weaknesses:
> **Weakness 1:** The deterministic nature of the STC-ViT model, which does not account for model uncertainty.
> - We have acknowledged the limitation of STC-ViT not accounting for uncertainties in our Conclusion section. While we cannot implement the suggested models right now, we highly appreciate your suggestions and consider extending STC-ViT to quantify uncertainties as future work.
>
> **Weakness 2:** There is a need for a more comprehensive discussion of the limitations and potential biases in the training data.
> - You are absolutely right about the limitations of using only one dataset. We address these concerns in more detail in Conclusion section. We acknowledge that the ERA5 dataset, while a robust and widely used reanalysis product, is not without limitations. Potential biases in data-sparse regions, challenges in representing local phenomena, and inconsistencies in observational continuity may impact model generalizability. To address this, future work will explore:
>    - Data Augmentation to synthetically increase dataset diversity.
>    - Integration with Diverse Datasets, such as regional or event-specific datasets, to mitigate geographical and climatic biases.
>
> **Weakness 3:** The authors could enhance the interpretability of the model by providing deeper insights into the attention mechanisms employed within STC-ViT.
> - We have refined the methodology section (see equation 19) and added sub diagram (see figure 1) to understand the structure of the model more clearly. We have also added the images of prediction results and biases in the appendix section B to understand where the model might not be performing very well.

---

> > ### Author Response · Authors · 2024-12-05
> >
> > ### Requested Changes:
> > **Refine the methodology section:**
> > - In response to weakness 3, we have refined the methodology section in following ways:
> >    - Added a sub-diagram (refer to figure 1) to explain the MLP structure of the Neural ODE layers
> >    - Refine the notations and equations for MHA to better understand the calculation of spatial and temporal attention
> >    - We have added an additional equation to explain the structure of how latent representation of attention is passed to the neural ode layers
> > \begin{equation}
> > h_{n+1}=h_n+ODEBlock(LayerNorm(Attention(h_n)))+ODEBlock(LayerNorm(MLP(h_n)))
> > \end{equation}
> >
> > **Include additional metrics:**
> > Thank you for your suggestion of including additional metrics. While our submission not only considers RMSE, but also Anomaly Correlation Coefficient (ACC), which accounts for prediction anomalies in reference to Climatology, we have now added mean absolute error as well in our results section of Appendix B (refer to table 5).  Further we have also conducted studies at longer lead times of 2 weeks to 8 weeks to assess the model’s prediction ability at sub seasonal forecasting (refer to table 4).
> >
> > | Variable| 6 hr  | 12 hr | 18 hr | 1 day | 3 day  | 6 day  | 2 week | 4 week | 6 week |
> > |---------|-------|-------|-------|-------|--------|--------|--------|--------|--------|
> > | z500(m^2/s^2)| 60.33 | 72.33 | 84.79 | 98.67 | 351.34 | 519.87 | 529.97 | 557.38 | 722.47 |
> > | u10 (m/s) | 0.65  | 0.79  | 0.88  | 1.00  | 2.30   | 2.83   | 2.85   | 2.92   | 3.11   |
> > | v10 (m/s)| 0.67  | 0.81  | 0.91  | 1.03  | 2.36   | 2.89   | 2.92   | 2.99   | 3.16   |
> > | T2m (K) | 0.61  | 0.72  | 0.78  | 0.81  | 1.44   | 2.26   | 2.39   | 2.77   | 4.15   |
> > | T850 (K)| 0.60  | 0.726 | 0.79  | 0.86  | 1.81   | 2.53   | 2.63   | 2.90   | 3.96   |
> >
> >
> > | **Variable**   | **2 weeks** | **4 weeks** | **6 weeks** | **8 weeks** |
> > |----------------|-------------|-------------|-------------|-------------|
> > | z500 (m^2/s^2) | 825.68      | 857.09      | 968.5       | 1068.5      |
> > | T2m (k)        | 3.68        | 4.31        | 5.24        | 6.35        |
> > | T850 (k)       | 3.67        | 4.05        | 4.82        | 5.54        |
> > | u10 (m/s)      | 4.00        | 4.05        | 4.23        | 4.45        |
> > | v10 (m/s)      | 4.07        | 4.13        | 4.28        | 4.42        |
> >
> > **Ablation Studies:**
> > We initially performed the ablation studies to understand the role of each component of STC-ViT. The results show that learning the latent representation from the continuous attention plays a major role in the prediction accuracy of the model.
> > We have now further added the ablation studies for each loss function showing that Thermodynamic loss has the highest influence on the prediction results of STC-ViT. (refer to section 5.5.2)
> >
> > **Computational Comparison:**
> > NWP models simulate the behavior of the atmosphere by solving differential equations at high spatial and temporal resolutions. This involves massive computations, real-time data assimilation, and iterative numerical methods, all of which demand the power and efficiency of supercomputers to generate accurate and timely forecasts which cannot be directly compared with data-driven AI based methods. As a comparison to other SOTA weather models we have added the run time details in the table 2
> >
> > | **Model**          | **Parameters** | **Training Time** | **Train Device**  |
> > |-----------------------------|-------------------------|----------------------------|----------------------------|
> > | PanguWeather                | 256M                    | 64 days                    | 192 NVIDIA Tesla-V100 GPUs |
> > | GraphCast                   | 37M                     | 4 weeks                    | 32 TPUs                    |
> > | ViT (ClimaX-non pretrained) | 107M                    | ~2 days                    | 8 V100                     |
> > | STC-ViT (5.625/1.5 degree)  | 98M                     | 2days/20days               | 4 V100/2 A100              |
> >
> >
> > **Societal Impact:**
> > To address your concern regarding the Societal impacts of the model we have expanded this section as follows:
> > Additionally, training STC-ViT at higher resolution on more diverse datasets and focusing on high resolution regional forecasts as a future work can directly contribute to enhanced climate resilience, enabling societies to better anticipate and adapt to extreme weather events such as hurricanes, droughts, and floods. Further, extending the study to cater for longer lead times of few months and seasonal can help in disaster preparedness, as timely and precise forecasts allow governments and organizations to implement early warning systems, plan evacuations, and allocate resources effectively, thereby mitigating loss of life and property.

---

### Review · Reviewer_SYMe · 2024-11-25

**Summary Of Contributions:**

The authors introduce several architectural and training approaches for the problem of data-driven global weather forecasting. The proposed model applies neural ODEs to the output of the sub-components of a transformer block which are applied before adding the result back into the residual stream. The authors also propose using an augmented training loss which includes physically inspired terms related to various conservation laws.

**Audience:**

No

**Broader Impact Concerns:**

None.

**Claims And Evidence:**

No

**Requested Changes:**

Overall, a lot of the stated weaknesses are not* (edited for typo) dealbreakers if they are fairly and accurately described in the paper text itself. If the intended model contribution was the inclusion of these MLP blocks, then it should be stated that way. Right now there's a lot of language that does not appear to be accurate.

**Critical**
1. Ensure the description of the algorithm matches what was performed in your studies. Someone should be reproduce the results from the text, but right now these seem far enough apart that this isn't going to happen.
2. Clean up the language - if you can't evaluate at arbitrary time points and get physically results, it's not a continuous time model. It's fine for imperfect physically-inspired losses to be useful, but then you need to explain that they are not the actual physics.
2. Include comparisons of run times or FLOPs when presenting model results. Ideally try to scale all architectures to be on the same playing field, but it's understandable that this isn't always possible.
3. Be clear on the limitations of this approach. If the current models required 2 DGX A100 80 GB pods, it's unlikely that they can scale to practical resolutions and this needs to be noted as an exploratory study.
4. Clean up the notation - use a consistent indexing system. Switching between time and space sub-indices is unnecessarily confusing. Use one indexing scheme and hold certain indices constant to denote when you aren't transforming over them. Define your coordinate system and state variables using different symbols.
5. Show more pictures even if this has to be in the appendix. One of the known weaknesses of this domain is that ML models can often get good numbers with significantly overblurred solutions. People will not trust the model without seeing actual forecasts. This can be in the appendix if necessary.

**Strongly recommended**
1. Run experiments at higher resolution. Most 1440x721 models are trained using 80GB cards. Currently, a grid cell in the lower resolution experiments amounts to more than half the UK. The higher resolution still covers the entire greater London area. Low resolution is a useful test bed for proof of concepts, but it avoids many of the challenges of handling high resolution data and the two problem settings are quite different to the point that comparisons between architectures intended for these resolutions aren't truly fair fair. If you can't run at higher resolution, then I'd recommend incorporating some longer run forecast metrics - at least sub-seasonal forecasting - as these are areas where lower resolution is still useful. This is a largely empirical study so it's hard to say it will be interesting to readers if it's working with a toy version of the problem.
2. Include the various losses in the ablations.
3. Add diagrams for your architectural choices to make it clearer what is actually going on.
4. Similarly include model size information across the comparisons.

**Strengths And Weaknesses:**

**Strengths:**

1. The background materials are well described and cleanly written.
2. The presentation of the figures and tables are very good.
3. Using physics-inspired methods to improve the reliability of physics prediction tasks is an important area worth significant study.

**Weaknesses:**

Maybe there's an error in the provided code that could clear this up, but right now I have some major concerns about correctness and clarity due to some differences between what is described and what is in the code. Here are a summary of these issues:
1. The methods described in the paper text are significantly different from the code.
    1.  Maybe this works out in how the batching is implemented, but TCA seems to be a difference in attention weights across the batch and head dimension rather than what is described in the paper. It's using the same base Q, K, V projections as the spatial attention as well so it appears space and time have the same underlying information which also doesn't match the paper. The lag in (BH) seems a bit strange in general since it seems like it would be possible to attend to different samples in the same batch, so I'm assuming this is structured in a way to be time.
    2. The text implies that the attention mechanism itself is continuous. In equation 18, it's stated that the attention function $h$ is being evaluated in continuous time. This is not the case. In the code, $h$ is evaluated once as a function of the input, then some function of that output is integrated forward in time.
2. This does not appear to actually be a continuous time method nor does it appear to approximate one as the code uses a single time step to pass from t=0 to t=1 in each layer. There is no way to sample intermediate states (on the intergrators) and no attempt to ensure convergence. Neural ODEs have seen use for both continuous time modeling and as an implicit depth model. This is closer to the latter, but in practice due to the fixed step size with a non-adaptive solver, it amounts to adding a a weight-tied 4-layer MLP with specific residual structure.
3. The physical losses introduced are overlooking a number of important physical properties of the system. It's possible they're effective reweighting mechanisms that facilitate learning, but currently they are not enforcing correct physics. In general, it's going to be very hard to enforce conservation given the system itself is open and so energy is entering and leaving via coupling with unobserved systems.
    1. In the text, it doesn't look like the physics-losses are weighted in any way. Apart from the latitude-weighting in RMSE which is likely the largest issue, density, volume, and material state equations are going to vary significantly between pressure surfaces which would impact conservation.
    2. Given the intended goal, the kinetic energy loss is not accurate as it does not include vertical velocity which is likely negligible at altitude, but is very important near the boundary layer.
    3. `z` in ERA5 is already `geopotential` in `m**2/s**2` rather than `geopotential_height` so the the field itself already reflects potential energy (at the given level of approximation).
    4. Thermodynamics are going to be very tricky as the stated equation does not account for the fact that this is an open system (which is admittedly very difficult to account for). Heat enters and leaves the atmosphere consistently. This is especially complicated near the boundary layer where there is also orographic forcing.
4. Fairness of comparisons and limitations
    1. The proposed architecture is essentially adding 4 MLP calls to each sub-block of a transformer (8 extra MLP evaluations per full block). The model itself in the 5 degree config only has 4 blocks so this is a significant increase. Currently there is no evaluation of the cost of these various models. Run-time is going to be the important one here as weight-tied architectures have fewer parameters but more function calls per parameter.
    2. With the listed compute, these studies should really be on higher resolution. Low-resolution forecasting is interesting from a climate perspective, but only medium-range weather forecasting is discussed here.
    3. Ablation details - No ablation on the loss terms. Are all models trained with the losses?
5. Notation is inconsistent and some choices really don't work. For instance, the subtext in equations 10-13 really isn't clear. I'd recommend swapping to $t$, $t-1$ or just use double subtext.

---

> ### Author Response · Authors · 2024-12-05
> **Response to Reviewer SYMe**
>
> Thank you for your thoughtful and detailed review of our submission. We greatly appreciate the time and effort you took to provide valuable feedback and constructive comments. Below, we address the weaknesses followed by the requested changes in detail.
>
> **Weakness 1 and 2:** The methods described in the paper text are significantly different from the code.
> - Thank you for pointing out the error which has been fixed now.
> - The continuous aspect lies in how the model uses an ODE block to evolve the attention outputs continuously over time. This integration allows the attention's influence to be modelled across a time interval rather than being limited to static computation. The ODE block ensures the attention's output evolves in a continuous manner over the time interval [0,1]. This allows the model to approximate the effects of continuous dynamics while using discrete computations for efficiency. The choice of a fixed time step is a design decision, balancing computational efficiency and complexity. Adaptive solvers or intermediate sampling could be explored in future work to better approximate continuous dynamics.
> - Even though the solver appears to process a single time step (t=0→t=1) it does so by evaluating the function at multiple points within that interval. A fixed time step solver (e.g., rk4) still evaluates the function multiple times within the interval, achieving a piecewise approximation of the continuous trajectory. Therefore, even with a fixed time step solver, the ODE block evaluates the function multiple times within the interval, capturing intermediate states and approximating the continuous trajectory.
>
> **Weakness 3:** The physical losses introduced are overlooking a number of important physical properties of the system.
> - We acknowledge that the system does not fully adhere to physical principles of the atmosphere. The physical losses in this framework are not designed to enforce hard constraints or strict conservation laws. Instead, they act as soft constraints to guide the learning process and help the model learn a reasonable approximation of these physical processes.
> - Kinetic energy: Thank you for pointing out the limitation in our design choice of loss function. Here, we are limited by the choice of dataset which does not have vertical velocity component; however, we aim to solve this issue in future by exploring data from multiple sources.
> - Potential energy: You are absolutely correct about the use of z here; we have noted the error and will fix it in our future experiments. however here it as a weighting mechanism and has the least affect on the model’s predictions as shown in the ablation study This loss acts as a soft guide to ensure the model's predictions of geopotential align closely with the true field, even in the presence of noisy data.
> - Thermodynamics: This loss approximates the principle of energy advection by balancing temporal and spatial temperature changes with wind advection. While it simplifies the system by excluding external heat fluxes and vertical dynamics, it effectively enforces a first-order approximation of energy transport in the atmosphere. These simplifications align with the goal of soft constraints: guiding learning rather than enforcing complete physical fidelity.
>
> **Weakness 4:** Fairness of comparisons and limitations
> - The addition of 4 MLP calls per sub-block and 8 per full block is a deliberate trade-off to incorporate the benefits of neural ODE-based modelling. While traditional transformers rely solely on stacked layers, the introduction of MLP layers tied to the ODE formulation enables continuous modeling of transformations and allows the model to simulate more nuanced dynamics over time.
> - To address your concerns regarding run times, we have included a table (refer to Table 2) of comparison on parameters and compute time with other models.
> - You are absolutely right that high-resolution predictions are critical for accurate weather forecasting. However, due to computational limitations we were unable to conduct our experiments on higher resolutions. The primary goal of our study was not to achieve state-of-the-art high-resolution forecasts but to investigate the potential improvements in forecasting capabilities enabled by introducing a continuous-depth model integrated with soft physical constraints. By focusing on this novel architectural approach, we aim to provide foundational insights that can later be extended to higher resolutions with more computational resources.
> - We have added the Ablation studies on physical losses to the text in section 5.5.2.
>
> **Weakness 5: Inconsistent notations:**
> - Thank you for your recommendation. We have made the necessary changes to the text.

---

> > ### Author Response · Authors · 2024-12-05
> >
> > ### Requested Changes(critical):
> > 1.	We have improved the explanation and description of our methodology to align with the fact that the model leverages Neural ODEs to capture smoother, continuous transformations. While this formulation aligns with continuous-depth modeling principles, the practical implementation focuses on fixed-interval evaluations rather than arbitrary time points.
> > 2.	We improve the section of Physics loss indicating that the resulting the loss function is inspired by physical principles, however it does not enforce strict physical laws but instead guides the model toward outputs that are consistent with physical expectations.
> > 3.	Since different ML based weather forecasting models are trained on different resolutions, it is not feasible to re-train each model on the same resolution. We have added a comparison table for several ML models trained on their original data.
> > | **Model**          | **Parameters** | **Training Time** | **Train Device**  |
> > |-----------------------------|-------------------------|----------------------------|----------------------------|
> > | PanguWeather                | 256M                    | 64 days                    | 192 NVIDIA Tesla-V100 GPUs |
> > | GraphCast                   | 37M                     | 4 weeks                    | 32 TPUs                    |
> > | ViT (ClimaX-non pretrained) | 107M                    | ~2 days                    | 8 V100                     |
> > | STC-ViT (5.625/1.5 degree)  | 98M                     | 2days/20days               | 4 V100/2 A100              |
> >
> > 4.	STC-ViT is trained on two resolutions 5.625 and 1.5 degree. Our lower resolution model is trained on 4 V100 32GB RAM and 1.5 degree is trained using 2 80GB A100. We acknowledge the computational complexity of using transformer model and discuss its limitation in the Conclusion section.
> > 5.	We have added a Notation section to provide clarity on the notations used in the paper and have tried to make them uniform across the paper. Throughout the paper t and t−1 is used to denote the information at current and previous time
> > step respectively. Vi(x, y, t) is used to denote weather variable with x and y dimensions at current time step t.
> > 6.	We have added the predicted forecasts and bias in the appendix B under qualitative results.
> >
> > ### Requested Changes(strongly recommended):
> > 1.	We acknowledge that low-resolution and high-resolution problems differ significantly in complexity, especially in capturing localized phenomena. The goal of this study is to evaluate the effectiveness of the proposed attention mechanism combined with soft physical constraints rather than direct high-resolution forecasting. While higher resolution experiments would undoubtedly add value, they are currently infeasible due to resource constraints. Instead, we have extended our study to include long-range forecasting up to 8 weeks.
> > | **Variable**   | **2 weeks** | **4 weeks** | **6 weeks** | **8 weeks** |
> > |----------------|-------------|-------------|-------------|-------------|
> > | z500 (m^2/s^2) | 825.68      | 857.09      | 968.5       | 1068.5      |
> > | T2m (k)        | 3.68        | 4.31        | 5.24        | 6.35        |
> > | T850 (k)       | 3.67        | 4.05        | 4.82        | 5.54        |
> > | u10 (m/s)      | 4.00        | 4.05        | 4.23        | 4.45        |
> > | v10 (m/s)      | 4.07        | 4.13        | 4.28        | 4.42        |
> >
> > 2.	We have included the ablation studies for different loss functions (refer to section 5.5.2)
> > 3.	We have added a sub-diagram to show the architecture of Neural ODE model (refer to figure 1)
> > 4.	We have included the model sizes in Table 2

---

### Author Response · Authors · 2024-12-05
**Key Revision Highlights**

We appreciate the invaluable feedback of all reviewers and we've incorporated your suggestions into our manuscript. Below are the key revisions made:
- **update methodology section:** We have updated the 'neural ode integration' of our methodology to make more clearer.
- **update use of potential energy:** We have corrected the usage of Potential Energy loss as a weighting mechanism rather than actual Physical law
- **physics Informed loss ablation studies:** We have performed ablation studies for each of the physics informed loss function
- **run-time comparison against SOTA:** We have added a table of comparison for training times of different data-driven models
- **discussion on limitations:** Expanded on the limitations of the current approach
- **additional metrics:** We computed additional metric of Mean Absolute Error and added in the appendix section B.1
- **sub seasonal results:** We computed additional metrics from 2 weeks to 8 weeks and added in the appendix section B.1
- **prediction maps:** We have added qualitative forecast results in the appendix section B.2

---

### Decision · Action_Editor_TaMU · 2025-01-19

**Recommendation:** Reject

**Comment:**

The paper introduces the Spatio-Temporal Continuous Vision Transformer (STC-ViT) for weather forecasting. The authors claim two main contributions: the integration of a continuous NeuralODE-based mechanism in the transformer’s attention to account for complex dynamics, and the use of physics-informed losses to enhance prediction accuracy.

The paper received mixed initial feedback. The main concerns raised by the reviewers included the clarity of the paper and the formalization of the proposed approach, the validity of the model's claimed continuous nature, discrepancies between the paper’s description and the code, and the compliance of the proposed physics-informed losses in the context of weather forecasting. Additionally, reviewers found that the experiments lacked fair comparisons with respect to baselines, particularly regarding low- and high-resolution results and the computational demands of the methods. While Reviewer C1mc voted for acceptance, Reviewer SYMe was not fully satisfied with the authors’ responses, considering that the inaccuracies in the claims and the insufficiencies in the experiments were not properly addressed. Reviewer 1yfU recommended weak acceptance, acknowledging the limitations highlighted by the other reviewers concerning the code and experimental validations.

The AE has thoroughly reviewed the submission and the discussion. The AE acknowledges that the paper addresses an important and timely problem. However, the AE believes that significant improvements are necessary to strengthen the current submission. Since the approach integrates standard components (NeuralODE, physics-informed losses applied to weather forecasting), the contributions should be clearly and fairly presented, and the experiments require robust validations to provide meaningful value to the community. Specifically, the AE considers that the implementation of the NeuralODE layers should be discussed in greater depth, particularly by clarifying what the proposed NeuralODE with fixed-time evaluation adds compared to a standard transformer with finer time sampling. Additionally, fair comparisons with baselines trained at higher resolutions should be included, addressing both accuracy and efficiency. The generalization properties of the physics-informed losses should also be analyzed more comprehensively. For these reasons, the AE recommends rejection.

**Audience:**

The paper addresses the problem of weather forecasting using physics-informed transformers, a topic of significant interest to the broad TMLR audience.

**Claims And Evidence:**

The claims made in the paper should be better supported with evidence, including a more thorough analysis of the proposed approach and fair comparisons with baselines in terms of accuracy and efficiency.

**Resubmission Of Major Revision:**

The authors may consider submitting a major revision at a later time.